# Mitigating Discretization Bias in Neural Stochastic Differential Equations with Inference-Time Dropout

## Abstract

Neural stochastic differential equations (NSDEs) provide a powerful framework for modeling complex continuous-time dynamics by combining deep learning with Itô calculus. However, prior work has largely overlooked a key source of error: the discretization bias introduced by the Euler–Maruyama scheme. We argue that this bias is intrinsic and persists regardless of whether the driving noise is Brownian motion or a more general process. To address this issue, we introduce a lightweight mechanism—inference-time dropout—which acts as a stochastic correction layer that counteracts discretization-induced errors during simulation and prediction. This mechanism also reframes dropout, showing its potential as a tool for modeling distributional uncertainty rather than only as a training-time regularizer. We provide a formal analysis and empirical results demonstrating that this approach improves the robustness of Neural SDEs across diverse stochastic settings. To facilitate further research, we release the code and implementation details at the following link: `https://anonymous.4open.science/r/FreeSDE-FC37/`.

## 1 Introduction

Stochastic differential equations (SDEs) describe how random processes evolve in continuous time. They have been widely applied in finance, climate science, and biology (Black and Scholes, 1973; Merton, 1973; Yang et al., 2020; Mariani et al., 2022; Alshammari and Khan, 2021; Kamrujjaman et al., 2022; Floris, 2015; Mishi et al., 2020). Neural stochastic differential equations (NSDEs) (Tzen and Raginsky, 2019) extend this classical framework by parameterizing drift and diffusion terms with neural networks. This combination enables data-driven learning of stochastic dynamics while preserving the formal structure of SDEs. Unlike discrete-time models, NSDEs naturally handle irregular sampling and provide principled uncertainty representations, making them attractive for real-world systems with complex temporal behaviors (Cohen et al., 2023; Gierjatowicz et al., 2022b; 2020; Cuchiero et al., 2024; Hwang et al., 2021; Giebel and Rainer, 2011).

Despite their promise, current NSDE methods inherit a critical limitation from numerical simulation. The commonly used Euler–Maruyama scheme introduces discretization bias: once a continuous-time process is approximated with discrete steps, its randomness no longer strictly follows the Gaussian law assumed by the scheme. This discrepancy becomes especially pronounced when the diffusion coefficient is nonlinear. As a result, the simulated dynamics systematically deviate from the true process, and the bias accumulates over long horizons. Importantly, this limitation persists not only for Brownian motion but also for more general Lévy processes. In Section 3.1, we formally analyze this phenomenon and show that the resulting error is intrinsic and cannot be ignored.

To address this issue, we introduce *inference-time dropout* as a lightweight, plug-and-play stochastic correction mechanism. By injecting Bernoulli-driven noise at prediction, it compensates for the Gaussian approximation implicit in Euler–Maruyama and expands the class of noise behaviors that NSDEs can capture. This reframes dropout from a training-only regularizer into a principled tool for uncertainty modeling in continuous-time systems. Our contributions are threefold:

- We introduce inference-time dropout as a lightweight, plug-and-play mechanism to mitigate discretization-induced errors in NSDEs.

- We provide a theoretical analysis showing that inference-time dropout can approximate a broad class of non-Gaussian noise distributions, thereby extending the expressiveness of NSDEs.
- We validate the approach on simulated and real-world tasks, demonstrating competitive performance in long-horizon forecasting and robustness of uncertainty quantification.

## 2 RELATED WORK

We review related work on NSDEs from two complementary perspectives: *(i) methodological advances*, focusing on training paradigms, *(ii) applications*, highlighting the use of NSDEs in diverse domains. Importantly, across both lines of work, the vast majority of methods discretize dynamics using the EM scheme, which introduces biases that remain largely unaddressed.

**Methodological Advances.** NSDEs model complex temporal dynamics in continuous time, but training involves balancing stochastic trajectory simulation with tractable gradient estimation. Early methods used variational inference for latent SDEs (Tzen and Raginsky, 2019; Li et al., 2020) and adversarial training to match path distributions (Kidger et al., 2021a). While GAN-based approaches can capture complex data distributions, they often suffer from instability. To address this, recent work has proposed non-adversarial paradigms, such as using signature kernel scores (Issa et al., 2023), which provide theoretical guarantees and mitigate mode collapse. Advances in gradient computation have also been crucial: the adjoint sensitivity method (Li et al., 2020) and reversible solvers (Kidger et al., 2021b) improve both accuracy and efficiency, enabling scalable training of NSDEs. Nevertheless, all of these approaches still rely on EM discretization as the numerical backbone.

**Applications.** NSDEs have been increasingly applied to stochastic continuous processes across domains. In finance, they are used for irregular time series forecasting (Oh et al., 2024), modeling asset dynamics, derivative pricing, and risk management (Gierjatowicz et al., 2022a; Kidger et al., 2021a). Beyond finance, NSDEs have been applied to general sequential data such as motion capture (Li et al., 2020), hybrid systems combining continuous flows with discrete jumps (e.g., temporal point processes (Jia and Benson, 2019)), and principled uncertainty quantification (Kong et al., 2020), broadening their role in decision-making and scientific modeling. Again, these applications inherit EM discretization as the default numerical scheme, and thus share the same bias limitations. Our work departs from the above by questioning this discretization choice and proposing a correction mechanism.

## 3 METHODOLOGY

We begin in Section 3.1 with an analysis of discretization bias, highlighting why the Euler–Maruyama scheme introduces systematic deviations in NSDE simulation. In Section 3.2, we present inference-time dropout as a lightweight correction mechanism. Section 3.3 then provides intuition for how inference-time dropout mitigates discretization-induced errors and expands the expressive power of NSDEs beyond Gaussian approximations. Finally, in Section 3.4, we offer a formal theoretical analysis that characterizes the approximation capacity of this mechanism, before moving on to empirical validation.

### 3.1 DISCRETIZATION BIAS ANALYSIS

**Preliminaries.** An NSDE is defined by parameterizing the drift and diffusion terms of an Itô SDE with neural networks:

$$dX_t = \mu_\theta(X_t, t)\, dt + \sigma_\phi(X_t, t)\, dW_t,$$

where $\mu_\theta$ and $\sigma_\phi$ are learnable neural networks and $W_t$ is a standard Wiener process. Training and simulation of NSDEs require discretization of this continuous-time system. The most widely used method is the EM scheme, which approximates the dynamics as

$$X_{k+1} = X_k + \mu_\theta(X_k, t_k)\, \Delta + \sigma_\phi(X_k, t_k)\, \Delta W_k, \quad \Delta W_k \sim \mathcal{N}(0, \Delta I).$$

This scheme is attractive due to its simplicity and low computational cost, and thus has become the de facto standard in existing NSDE works.

**Discretization Bias.** The Euler–Maruyama (EM) scheme, though widely adopted, suffers from an *unavoidable* discretization bias. Even under global Lipschitz and linear growth conditions, Theorem 3.1 shows that EM achieves only strong order $1/2$ and weak order $1$. This implies two fundamental limitations: (*i*) an order barrier that prevents eliminating bias even with small step sizes, and (*ii*) error accumulation over long horizons due to the exponential dependence on $T$. Consequently, NSDEs built upon EM discretization inevitably inherit this bias, leading to systematic distortion of the noise distribution when the diffusion coefficient is nonlinear, and degrading both predictive performance and uncertainty quantification.

**Theorem 3.1** (Error bounds for Euler–Maruyama). *Consider the Itô SDE*

$$dX_t = \mu(X_t, t)\, dt + \sigma(X_t, t)\, dW_t, \quad X_0 \in \mathbb{R}^d,$$

*where $\mu$ and $\sigma$ satisfy global Lipschitz and linear growth conditions. Let $X_T$ denote the exact solution at time $T$ and $X_T^\Delta$ the Euler–Maruyama approximation with step size $\Delta$. Then there exist constants $C, C_f > 0$ depending on $\mu, \sigma, f$ such that:*

$$\textit{(Strong error)} \quad \left(\mathbb{E}\|X_T - X_T^\Delta\|^2\right)^{1/2} \leq C\, e^{CT}\, \Delta^{1/2}, \tag{1}$$

$$\textit{(Weak error)} \quad \left|\mathbb{E}[f(X_T)] - \mathbb{E}[f(X_T^\Delta)]\right| \leq C_f\, e^{CT}\, \Delta, \tag{2}$$

*for any $f \in C_p^4(\mathbb{R}^d)$ (polynomially bounded smooth test function).*

*Proof.* The proof follows the standard recursion argument. Let $e_k = X_{t_k} - X_k^\Delta$ denote the grid-point error. Subtracting the EM update from the exact mild solution yields

$$e_{k+1} = e_k + \int_{t_k}^{t_{k+1}} (\mu(X_s, s) - \mu(X_k^\Delta, t_k))\, ds + \int_{t_k}^{t_{k+1}} (\sigma(X_s, s) - \sigma(X_k^\Delta, t_k))\, dW_s.$$

Applying Itô isometry, Lipschitz bounds, and moment estimates gives the discrete recursion

$$\mathbb{E}\|e_{k+1}\|^2 \leq (1 + C\Delta)\, \mathbb{E}\|e_k\|^2 + C\Delta^2.$$

A discrete Grönwall inequality then yields

$$\mathbb{E}\|e_N\|^2 \leq C e^{CT} \Delta,$$

establishing the strong error bound equation 1. For the weak error, consider $u(t, x) = \mathbb{E}[f(X_T) \,|\, X_t = x]$, which solves the Kolmogorov backward PDE. A Talay–Tubaro expansion shows that the weak error admits an asymptotic expansion in $\Delta$ with leading term $O(\Delta)$, and constants growing as $e^{CT}$, giving equation 2. $\qquad\square$

### 3.2 Inference-Time Dropout

**Inference-Time Dropout Implements.** Motivated by the limitations highlighted in Theorem 3.1, we introduce *Inference-Time Dropout* (ITD) as a stochastic correction mechanism at the discretization level. The key idea is to preserve dropout masks during inference and repurpose them as a source of randomness, thereby replacing the Gaussian increments assumed in EM. Concretely, we propose to update the state as

$$X_{k+1} = X_k + \mu_\theta(X_k, t_k)\, \Delta t + \sigma_\phi(X_k, t_k)\, \epsilon_\theta(\Delta t),$$

where $\epsilon_\theta(\Delta t)$ is a random vector generated by a neural network with fixed dropout masks. Unlike $\Delta W_k$, whose distribution is restricted to Gaussian, the law of $\epsilon_\theta(\Delta t)$ is implicitly defined by the dropout pattern and network parameters, allowing the model to approximate richer, non-Gaussian increment distributions.

In this way, ITD directly intervenes in the discretization step: it substitutes the Gaussian surrogate in EM with dropout-induced randomness, effectively reshaping the distribution of discrete-time updates. This preserves the computational simplicity of EM while mitigating its inherent distributional mismatch. Moreover, ITD can be extended hierarchically to capture stochasticity across temporal scales. Parallel branches with different dropout rates ($p_1 < p_2 < p_3$) generate complementary noise components—ranging from high-frequency fluctuations to low-frequency trends. A soft orthogonality constraint across branches encourages diversity, ensuring that the induced increments approximate a broad class of non-Gaussian behaviors.

### 3.3 Understanding Inference-Time Dropout

**Connection to Mixture-of-Experts.** ITD can be viewed as a generalized form of Mixture-of-Experts (MoE) (**?**). For a given input $x$, let $M$ denote the dropout mask and $f_\theta(M, x)$ the corresponding network output. The distribution induced by dropout can be written as $p(y \mid x) = \sum_M p(M)\, \delta_{f_\theta(M,x)}(y)$, where $p(M)$ is the Bernoulli probability of selecting mask $M$. This corresponds exactly to a mixture model in which each "expert" is a subnetwork $f_\theta(M, \cdot)$ and the routing distribution is fixed by the dropout probabilities. In standard MoE, the output distribution takes the form $p(y \mid x) = \sum_{i=1}^K \pi_i(x)\, \delta_{f_i(x)}(y)$, where gating weights $\pi_i(x)$ adaptively depend on the input. By contrast, ITD replaces adaptive gating with stochastic but static routing: $\pi_i(x) = p(M_i)$, independent of $x$. If dropout is applied only to the final layer, the mechanism degenerates to a fixed router plus several subnetworks that share lower-layer parameters: $p(y \mid x) = \sum_{j=1}^N p_j\, \delta_{g_j(x)}(y)$, which can be interpreted as a simplified MoE with a shared backbone and $N$ experts $g_j$. When dropout is applied across multiple layers, the effective distribution becomes conditional on $x$, as the outputs $f_\theta(M, x)$ vary with both input and dropout mask, yielding a richer conditional mixture family.

**Connection to Gaussian Mixtures.** ITD also shares structural similarities with mixture models such as Gaussian Mixture Models (GMMs). Formally, the distribution induced by ITD can be written as $\sum_i p_i \delta_{f_\theta(M_i, x)}$, where each atom corresponds to the deterministic output of a subnetwork and $\{p_i\}$ are determined by dropout probabilities. This is analogous to GMMs $\sum_j \alpha_j \mathcal{N}(\mu_j, \Sigma_j)$, where mixture richness comes from combining multiple Gaussian components. The key difference lies in flexibility: GMMs use fixed kernel families, whereas ITD generates data-adaptive atoms through neural subnetworks. Moreover, convolving the discrete dropout mixture with a small Gaussian kernel yields a continuous approximation, effectively bridging dropout-induced distributions with classical mixture models.

### 3.4 Theoretical Analysis

We now develop a theoretical framework that justifies interpreting ITD as a universal sampling mechanism, capable of approximating any conditional output distribution through appropriate network design. The key idea is that injecting independent Bernoulli masks into the hidden units at inference time induces a randomized computation graph whose law corresponds to sampling from an explicit distribution over networks. This view naturally connects dropout to Bayesian reasoning: each forward pass can be seen as a Monte Carlo sample from a posterior-like ensemble. The remainder of this section is devoted to formalizing this equivalence and quantifying the approximation guarantees it entails.

**Formulations.** Let $X \subset \mathbb{R}^{d_x}$ be compact and $Y \subset \mathbb{R}^{d_y}$ be bounded. For each $x \in X$, suppose we are given a target conditional distribution $P_{Y|X=x}$ on $Y$. Fix a dropout rate $p \in (0,1)$, a network depth $L \geq 1$, and widths $(n_\ell)_{\ell=1}^{L-1}$. For every input $x$, we define a *random* feed-forward ReLU network with dropout:

$$f_\theta(M, x) = \big(\phi_L \circ D_{M^{(L-1)}} \circ \phi_{L-1} \circ \cdots \circ D_{M^{(1)}} \circ \phi_1\big)(x), \tag{3}$$

where $\theta$ denotes all deterministic weights and biases, and each $M^{(\ell)} \sim \text{Ber}(p)^{n_\ell}$ is an independent Bernoulli mask sampled *at inference time*, applied via the diagonal operator $D_{M^{(\ell)}}$. Each $\phi_\ell$ represents the affine transformation of layer $\ell$ followed by ReLU activation.

We denote by $\mathcal{L}(f_\theta(M, x))$ the distribution of the network output induced by the randomness in $M$. Discrepancies between distributions are measured using the 1-Wasserstein metric $\mathcal{W}_1$ on the bounded space $Y$.

The following regularity conditions are assumed for the target conditional distribution:

**Assumption 3.2.** The map $x \mapsto P_{Y|X=x}$ is weakly continuous on $X$.

**Assumption 3.3.** For every $x \in X$, the distribution $P_{Y|X=x}$ admits a continuous cumulative distribution function (CDF) in each coordinate; equivalently, the associated quantile function $F_x^{-1} : (0,1) \to Y$ is continuous.

We are now ready to state the main result of this section, which establishes that inference-time dropout can universally approximate conditional distributions in Wasserstein distance.

**Theorem 3.4** (Universal Approximation with Inference-Time Dropout). *Assume that Conditions 3.2 and 3.3 hold. Then for any $\varepsilon > 0$, there exist a depth $L$, widths $(n_\ell)$, dropout rate $p \in (0, 1)$, and weights $\theta$ such that the network defined in equation 3 satisfies*

$$\sup_{x \in X} \mathcal{W}_1\big(\mathcal{L}(f_\theta(M, x)), \, P_{Y|X=x}\big) < \varepsilon.$$

Theorem 3.4 formalizes the expressive power of inference-time dropout: under mild continuity assumptions, dropout-induced random networks can approximate any target conditional distribution to arbitrary accuracy in $\mathcal{W}_1$ distance. This result provides a rigorous basis for viewing dropout as a mechanism for posterior approximation and supports its use in uncertainty quantification tasks.

The proof of Theorem 3.4 is constructive: we first approximate the target distribution via quantile functions and then construct a randomized network whose output distribution matches it. Full details are deferred to Appendix 4.

## 4 EXPERIMENTS

In Section 4.1, we first describe the datasets, baselines, and evaluation metrics. Section 4.2 then reports the main simulation results under a range of synthetic and semi-synthetic settings. Section 4.3 examines the empirical performance of our approach on three practical tasks—reconstruction, prediction, and uncertainty estimation. Finally, additional details are provided in the Appendix.

### 4.1 EXPERIMENTS SETTING

**Simulated Datasets and Real-world Datasets.** The simulated datasets are carefully designed to cover both Gaussian and non-Gaussian noise regimes. Specifically, Gaussian noise scenarios include classical models such as Geometric Brownian Motion, Ornstein-Uhlenbeck process, and Cox-Ingersoll-Ross process. Non-Gaussian cases are represented through jump diffusion processes with Poisson innovations and generalized Gamma processes with composite noise. The detailed parameter configurations for these data generation processes are summarized in Appendix 4. For the evaluation on real-world scenarios, we follow the protocol of (Park et al., 2021) and assess cross-domain performance on four datasets: PhysioNet clinical time series for medical monitoring, the Speech Commands audio corpus for voice recognition, Beijing Air Quality Index for environmental sensing, and S&P 500 tick data for financial markets.

**Baselines.** Our comparisons focus primarily on established SDE-based generative models, including Latent-SDE (Li et al., 2020) and GAN-SDE (Li et al., 2020), which provide the most direct benchmarks for evaluating our proposed method. To provide a broader perspective, we also include ODE-based approaches like ODE$^2$VAE (Yildiz et al., 2019) and Latent-ODE (Rubanova et al., 2019), together with strong sequence modeling baselines including GRU-D (Che et al., 2018) and mTAND (Shukla and Marlin, 2021).

**Metrics.** For the simulation studies, we evaluate the model's ability to recover the underlying DGP by measuring the maximum mean discrepancy (MMD) between the learned and true distributions. On real-world benchmarks, we focus on three core aspects: temporal forecasting accuracy, measured by mean squared error (MSE); distributional alignment, assessed via MMD between generated and observed sequences; and probabilistic calibration, evaluated through negative log-likelihood (NLL) on both prediction and reconstruction tasks.

### 4.2 SIMULATION RESULTS ANALYSIS

As shown in Table 1, models based on SDEs consistently outperform those built on ODEs and recurrent neural networks (RNNs) across all simulated processes. Incorporating ITD further yields order-of-magnitude improvements by mitigating discretization errors, with particularly strong gains on complex distributions. To better understand how ITD contributes to these improvements, Figure 1

| Model | OU | GBM | CIR | Jump | Gamma | Phys | PL | Poly |
|---|---|---|---|---|---|---|---|---|
| GRU-D | 4.40 | 2.16 | 12.79 | 2.66 | 60.11 | 16.12 | 2.79 | 3.50 |
| mTAND | 2.94 | **0.04** | 32.08 | 18.00 | 40.40 | 73.34 | 32.08 | 37.59 |
| ODE$^2$VAE | 3.50 | 5.50 | 15.11 | 27.55 | 26.62 | 152.45 | 15.11 | 103.21 |
| Latent-ODE | 16.97 | 7.50 | 12.41 | 23.50 | 43.74 | 65.45 | 12.41 | 4.78 |
| Latent-SDE | 3.39 | 9.55 | 12.36 | 8.57 | 31.20 | 42.07 | 9.36 | 3.51 |
| GAN-SDE | 5.75 | 9.88 | 11.61 | 8.91 | 35.46 | 40.87 | 11.26 | 6.51 |
| **Latent-SDE (w/ ITD)** | **0.61** (-82%) | **0.79** (-92%) | **0.98** (-92%) | **1.40** (-84%) | **0.55** (-98%) | **3.72** (-91%) | **1.12** (-88%) | **1.73** (-51%) |
| **GAN-SDE (w/ ITD)** | **0.60** (-90%) | **0.89** (-91%) | **0.15** (-99%) | **2.57** (-71%) | **0.96** (-97%) | **5.05** (-88%) | **2.09** (-81%) | **3.32** (-49%) |

Table 1: Performance comparison of models on simulated stochastic process generation tasks. All values are MMD $\times 10^{-2}$, rounded to two decimals. **Bold** indicates the best result per column. Red percentages denote the relative reduction compared to the corresponding baseline.

visualizes the model's ability to recover key stochastic properties. The top panels show that the generated noise increments align closely with the theoretical distributions for both Gaussian (GBM) and non-Gaussian (Jump) processes, validating the fidelity of the learned noise structure. The bottom panels illustrate the agreement between predicted and true terminal value distributions, highlighting the model's capacity to capture sequence-level distributions accurately.

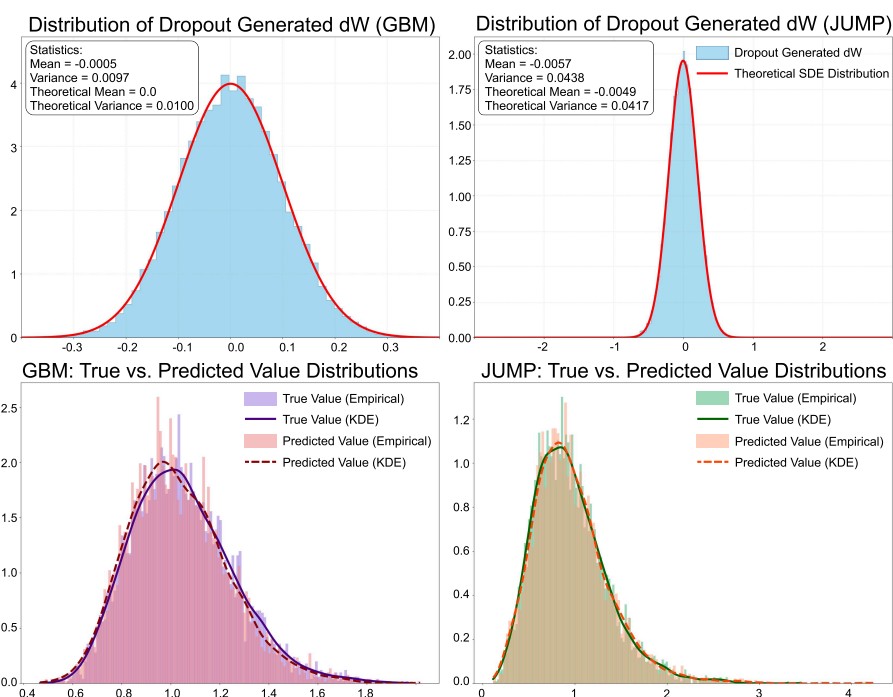

Figure 1: Visualization of the model's ability to recover stochastic properties using ITD. The top panels show that the generated noise increments align closely with theoretical distributions for both Gaussian (GBM) and non-Gaussian (Jump) processes, validating the fidelity of the learned noise structure. The bottom panels illustrate the agreement between predicted and true terminal value distributions, highlighting the model's capacity to capture sequence-level distributions accurately.

### 4.3 REAL-WORLD RESULTS ANALYSIS

The weaker performance of RNN- and ODE-based baselines can be attributed to their lack of explicit stochasticity and their limited ability to cope with non-stationary dynamics. These models tend to treat randomness as deterministic variation, which leads to systematic mis-specification in domains where uncertainty is intrinsic. As a consequence, they either overfit noise or drift away from the true dynamics in long-horizon prediction and reconstruction tasks, as reflected in Table 2. In contrast, by injecting structured randomness at inference, ITD restores the capacity of SDE-based models to represent stochastic fluctuations faithfully. This mechanism not only compensates for discretization

bias but also improves robustness and adaptability to real-world conditions where variability and non-stationarity are fundamental rather than incidental.

| Method | Reconstruction Tasks | | | | Prediction Tasks | | | |
|--------|------------|------|------|------|------------|------|------|------|
| | PhysioNet | | Speech Commands | | PhysioNet | | Air Quality | |
| | MSE | NLL | MSE | NLL | MSE | NLL | MSE | NLL |
| RNN-VAE | 6.66 | -33.24 | 0.98 | -4.92 | 6.23 | -25.81 | 6.72 | -4.84 |
| ODE$^2$-VAE | 5.52 | -24.20 | 0.72 | -3.76 | 5.37 | -26.81 | 7.94 | -3.97 |
| mTAND | 0.89 | -4.40 | 0.58 | -2.90 | 1.55 | -7.72 | **1.87** | -0.93 |
| Latent-ODE | 2.38 | -11.84 | 0.88 | -4.40 | 2.04 | -10.15 | 3.40 | -1.77 |
| Latent-SDE | 2.42 | -12.05 | 0.85 | -4.25 | 2.00 | -9.98 | 3.68 | -1.83 |
| GAN-SDE | 2.60 | -9.05 | 0.89 | -3.25 | 2.04 | -8.97 | 3.97 | -1.34 |
| **Latent-SDE (w/ ITD)** | **0.84** | **-0.10** | **0.26** | **-0.01** | **0.72** | **-5.35** | 2.30 | **-0.03** |

Table 2: Evaluation of reconstruction and prediction tasks. For readability, values are rescaled: MSE on PhysioNet is reported as $\times 10^{-1}$, and NLL as $\times 10^2$; MSE on Speech Commands is reported as $\times 10^0$ and NLL as $\times 10^3$; NLL on Air Quality as $\times 10^3$. **Bold** indicates the best result per column.

**Uncertainty Estimation** Uncertainty estimation is a core task in modeling dynamical systems with wide relevance in science, industry, and society. In financial time series, it is essential to capture distributional shifts, volatility, and tail risks such as black-swan events, which directly affect risk management, portfolio allocation, and algorithmic trading. Table 3 shows that adding ITD to Latent-SDE leads to a substantial improvement in uncertainty estimation, reducing MMD by orders of magnitude compared to all baselines. This demonstrates that ITD not only mitigates discretization error but also enables SDE-based models to capture non-Gaussian stochastic behavior that standard approaches fail to represent.

| Method | MMD |
|--------|-----|
| RNN-VAE | 427.05 |
| Latent ODE | 194.75 |
| Latent SDE | 190.11 |
| Latent-SDE (w/ ITD) | **0.27** |

Table 3: Uncertainty estimation on the Stock Market dataset (MMD $\times 10^{-3}$). **Bold** indicates the best result.

## 4.4 MORE ANALYSIS

**ITD for Distributional Approximation.** For a fixed input, ITD is equivalent to sampling from a distribution, since stochastic dropout masks induce random perturbations across forward passes, yielding an empirical distribution $q_\theta(z|x)$. Unlike reparameterization methods, ITD does not rely on explicit transformations of the sampling operator but instead exploits intrinsic network stochasticity. As shown in Figure 2, ITD reproduces a wide range of target distributions—including unimodal, heavy-tailed, skewed, bounded, and multimodal Gaussian mixtures—without requiring explicit mixture priors, confirming its effectiveness as an approximate differentiable sampling mechanism. To further validate this property in a generative setting, we conduct a controlled generation study. A decoder-only architecture is trained to map a simple Gaussian prior to the target distribution, and we compare three settings:

- **Input noise only**: dropout disabled, randomness introduced with resampling latent vectors.

- **Inference-time dropout (ITD)**: dropout activated during inference while keeping the latent vector fixed, injecting stochasticity inside the mapping.

- **Deterministic baseline**: both latent vector and dropout fixed, yielding identical outputs.

As shown in Figure 3, the ITD setting produces diverse outputs from a fixed latent vector, indicating that the model has learned to approximate high-dimensional distributions through internal stochasticity. In contrast, input-only noise confines randomness to the latent space, while the deterministic baseline collapses to identical outputs.

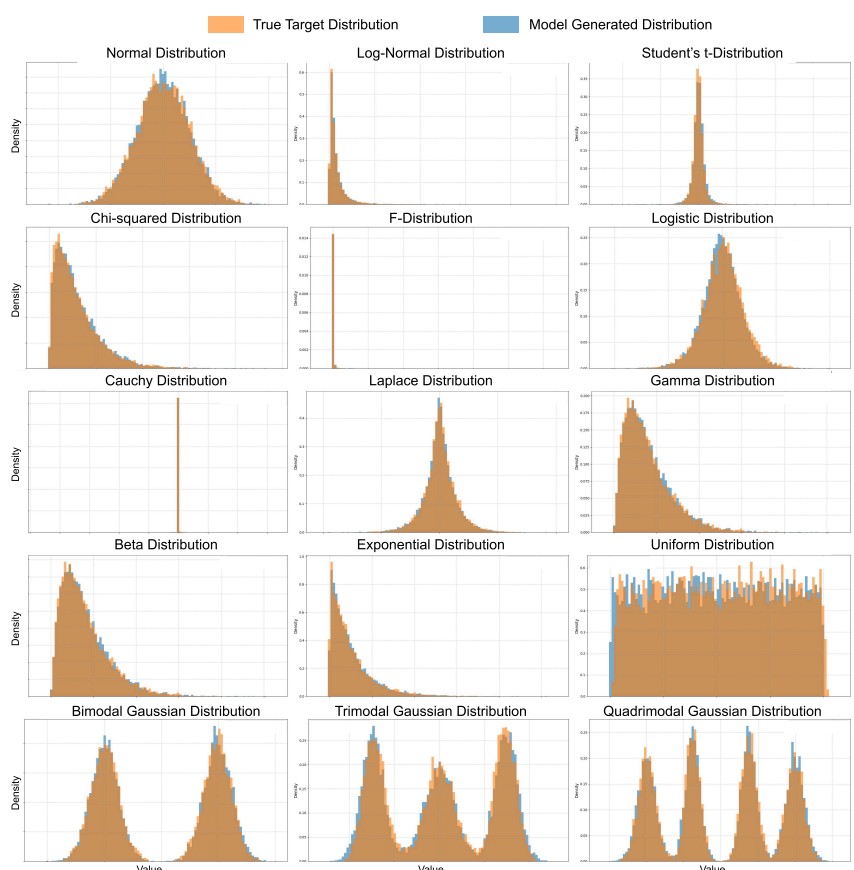

Figure 2: Each subplot compares the target distribution (orange) with the distribution induced by ITD (blue) under the same sampling scheme. The evaluation covers a broad class of distributions, including light- and heavy-tailed (e.g., Normal, Cauchy), skewed (e.g., Gamma, Exponential), bounded (e.g., Beta, Uniform), and multimodal cases (bimodal, trimodal, quadrimodal Gaussian mixtures). Detailed parameter configurations are provided in the Appendix 4.

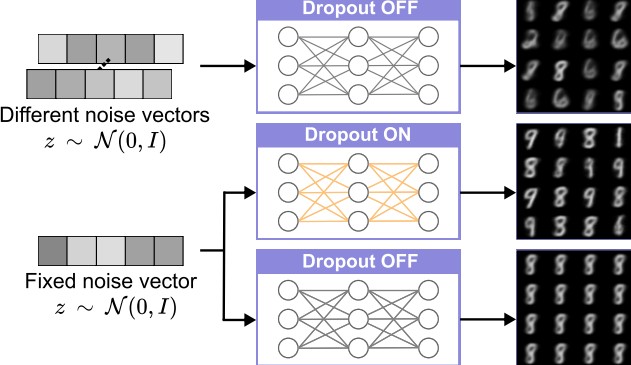

Figure 3: Visualization of generated digits under different stochasticity settings. Top: randomness injected via different latent vectors with dropout disabled. Middle: inference-time dropout with a fixed latent vector. Bottom: deterministic setting with both fixed latent vector and no dropout. The middle row exhibits the richest diversity, showing that inference-time dropout captures distributional variability through internal sampling. Detailed experimental configurations are provided in the Appendix 4.

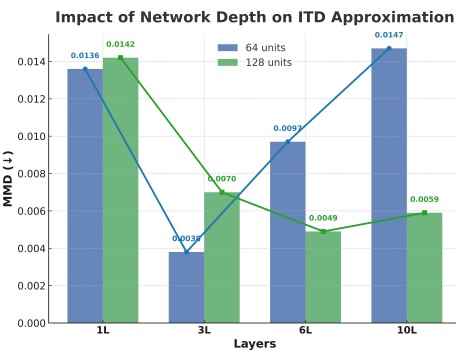

Figure 4: Impact of network depth on ITD approximation. Results are reported for models with 64 and 128 hidden units. Performance is evaluated using MMD ($\downarrow$) on the quadrimodal Gaussian benchmark.

**Impact of Dropout Configuration on Model Behavior.** To gain deeper insight into how ITD contributes to approximating distributions under discretization, we conduct a systematic hyperparameter study. The results indicate that **network depth has a non-monotonic effect**: shallow architectures underfit due to limited capacity, while very deep ones encounter optimization difficulties or excessive regularization. As shown in Figure 4, the most reliable trade-off emerges with intermediate depths of roughly three to six layers, which consistently achieve lower MMD values across different hidden-unit settings. This observation is consistent with our analysis in Section 3.2, where the effect can be viewed as implicitly learning $2^N$ subnetworks through dropout-based stochastic routing. When the number of neurons $N$ becomes too large, the resulting explosion in implicit models makes optimization substantially harder.

We observe that **increasing dropout coverage enhances distributional fitting**. As shown in Table 4, the lowest MMD is obtained when dropout is applied across all layers, indicating that injecting stochasticity throughout the network is most effective for capturing complex noise. If randomness is injected only in the final layers, each forward pass first traverses a deterministic backbone and then stochastically activates a subset of subnetworks through dropout masks, effectively forming an implicit ensemble of experts. This behavior is consistent with our earlier analysis in Section 3.3, where dropout was interpreted as a form of static stochastic routing. In this case, the mechanism degenerates into a simplified Mixture-of-Experts, which reduces the diversity of stochasticity and ultimately degrades model performance.

| Drop. Layers | MMD ($\times 10^{-3}$) |
|:---:|:---:|
| 1 | 16.2 |
| 3 | 12.9 |
| 6 | 22.3 |
| 10 | **5.9** |

Table 4: Effect of varying the number of *final* layers to which dropout is applied. All models use 10 layers with 128 hidden units per layer, and a fixed dropout rate $p = 0.5$. Performance is evaluated using MMD ($\downarrow$) on the quadrimodal Gaussian task. Values are reported in scientific notation, $\times 10^{-3}$; the best result is shown in bold.

## 5 LIMITATION

Our approach alleviates the discretization errors inherent in conventional NSDEs by introducing inference-time dropout as an adaptive source of stochasticity. While our theoretical and empirical results demonstrate clear advantages in predictive accuracy and uncertainty estimation, an important limitation lies in scalability. In particular, it remains unclear how well the framework extends to settings with massive datasets and very large parameter counts, such as those encountered in large language models. We leave the investigation of scaling laws and efficient training strategies for such large-scale deployments to future work.

## 6 CONCLUSION

We proposed ITD as a simple yet effective mechanism for enhancing neural SDEs. Theoretically, we established that ITD can be interpreted as a universal stochastic generator, capable of approximating arbitrary conditional distributions, and showed that it mitigates discretization errors inherent in NSDE formulations. Empirically, incorporating ITD into Latent SDEs improves both predictive accuracy and uncertainty estimation across synthetic and real-world datasets. These findings demonstrate that ITD provides a principled and practical tool for uncertainty-aware sequence modeling.

ETHICS STATEMENT AND REPRODUCIBILITY STATEMENT

This paper aims to advance the field of Machine Learning. While the work may have potential societal implications, we do not identify any specific ethical concerns that require special attention. We provide sufficient details of the model, training procedure, and evaluation setup to allow independent reproduction of our results. All hyperparameters, datasets, and experimental settings are documented in the paper or supplementary material.

REPRODUCIBILITY CHECKLIST

**Instructions for Authors:**

This document outlines key aspects for assessing reproducibility. Please provide your input by editing this .tex file directly.

For each question (that applies), replace the "Type your response here" text with your answer.

**Example:** If a question appears as

```
\question{Proofs of all novel claims are included}
{(yes/partial/no)}
Type your response here
```

you would change it to:

```
\question{Proofs of all novel claims are included}
{(yes/partial/no)}
yes
```

Please make sure to:

- Replace ONLY the "Type your response here" text and nothing else.

- Use one of the options listed for that question (e.g., **yes**, **no**, **partial**, or **NA**).

- **Not** modify any other part of the \question command or any other lines in this document.

You can \input this .tex file right before \end{document} of your main file or compile it as a stand-alone document. Check the instructions on your conference's website to see if you will be asked to provide this checklist with your paper or separately.

**1. General Paper Structure**

1.1. Includes a conceptual outline and/or pseudocode description of AI methods introduced (yes/partial/no/NA) yes

1.2. Clearly delineates statements that are opinions, hypothesis, and speculation from objective facts and results (yes/no) yes

1.3. Provides well-marked pedagogical references for less-familiar readers to gain background necessary to replicate the paper (yes/no) yes

**2. Theoretical Contributions**

2.1. Does this paper make theoretical contributions? (yes/no) yes

If yes, please address the following points:

2.2. All assumptions and restrictions are stated clearly and formally (yes/partial/no) yes

2.3. All novel claims are stated formally (e.g., in theorem statements) (yes/partial/no) yes

2.4. Proofs of all novel claims are included (yes/partial/no) yes

2.5. Proof sketches or intuitions are given for complex and/or novel results (yes/partial/no) yes

2.6. Appropriate citations to theoretical tools used are given (yes/partial/no) yes

2.7. All theoretical claims are demonstrated empirically to hold (yes/partial/no/NA) yes

2.8. All experimental code used to eliminate or disprove claims is included (yes/no/NA) yes

### 3. Dataset Usage

3.1. Does this paper rely on one or more datasets? (yes/no) Type your response here

If yes, please address the following points:

3.2. A motivation is given for why the experiments are conducted on the selected datasets (yes/partial/no/NA) yes

3.3. All novel datasets introduced in this paper are included in a data appendix (yes/partial/no/NA) yes

3.4. All novel datasets introduced in this paper will be made publicly available upon publication of the paper with a license that allows free usage for research purposes (yes/partial/no/NA) yes

3.5. All datasets drawn from the existing literature (potentially including authors' own previously published work) are accompanied by appropriate citations (yes/no/NA) yes

3.6. All datasets drawn from the existing literature (potentially including authors' own previously published work) are publicly available (yes/partial/no/NA) yes

3.7. All datasets that are not publicly available are described in detail, with explanation why publicly available alternatives are not scientifically satisficing (yes/partial/no/NA) NA

### 4. Computational Experiments

4.1. Does this paper include computational experiments? (yes/no) yes

If yes, please address the following points:

4.2. This paper states the number and range of values tried per (hyper-) parameter during development of the paper, along with the criterion used for selecting the final parameter setting (yes/partial/no/NA) yes

4.3. Any code required for pre-processing data is included in the appendix (yes/partial/no) yes

4.4. All source code required for conducting and analyzing the experiments is included in a code appendix (yes/partial/no) yes

4.5. All source code required for conducting and analyzing the experiments will be made publicly available upon publication of the paper with a license that allows free usage for research purposes (yes/partial/no) yes

4.6. All source code implementing new methods have comments detailing the implementation, with references to the paper where each step comes from (yes/partial/no) yes

4.7. If an algorithm depends on randomness, then the method used for setting seeds is described in a way sufficient to allow replication of results (yes/partial/no/NA) yes

4.8. This paper specifies the computing infrastructure used for running experiments (hardware and software), including GPU/CPU models; amount of memory; operating system; names and versions of relevant software libraries and frameworks (yes/partial/no) no

4.9. This paper formally describes evaluation metrics used and explains the motivation for choosing these metrics (yes/partial/no) yes

4.10. This paper states the number of algorithm runs used to compute each reported result (yes/no) yes

4.11. Analysis of experiments goes beyond single-dimensional summaries of performance (e.g., average; median) to include measures of variation, confidence, or other distributional information (yes/no) yes

4.12. The significance of any improvement or decrease in performance is judged using appropriate statistical tests (e.g., Wilcoxon signed-rank) (yes/partial/no) no

4.13. This paper lists all final (hyper-)parameters used for each model/algorithm in the paper's experiments (yes/partial/no/NA) yes

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

## EXPERIMENTAL SETUP

### SIMULATED DATASET CONFIGURATION

**Stochastic Process Configuration.** Our simulation datasets encompass two types of SDEs. The first type is designed to confirm that our approach maintains stable performance under the standard Gaussian noise setting. The second type, which is more challenging, involves non-Gaussian noise in SDEs. In the subsequent sections, we will delve into these two types of SDEs and detail their specific parameter configurations. We begin by examining a class of SDEs driven by standard Gaussian noise. These serve as controlled environments for evaluating the stability and consistency of our approach under well-understood dynamics. In particular, many of these models admit closed-form or well-characterized solutions, making them ideal for benchmarking and diagnostic purposes. The most canonical example is the Geometric Brownian Motion (GBM), which is analytically solvable via Itô's lemma and is widely used in mathematical finance and physics.

Typically, the Geometric Brownian Motion is analytically tractable due to its explicit closed-form solution derived via Itô's lemma. The process is governed by the stochastic differential equation: $dX_t = \mu X_t \, dt + \sigma X_t \, dW_t$ with the solution expressed as: $X_t = X_0 \exp\left(\left(\mu - \frac{\sigma^2}{2}\right)t + \sigma W_t\right)$ where the exponential transformation guarantees path positivity. The SDEs in this group share the common structure of being driven by Brownian motion, $dW_t$, and include both linear and nonlinear dynamics. Their formulations and parameter detailed configurations are listed below.

- **Geometric Brownian Motion (GBM).** *Formula*: $dX_t = \mu X_t dt + \sigma X_t dW_t$. *Parameters*: $\mu = 0.05$, $\sigma = 0.2$, $x_0 = 1.0$, $T = 1.0$, $n\_steps = 100$.

- **Ornstein-Uhlenbeck Process (OU).** *Formula*: $dX_t = \kappa(\alpha - X_t)dt + \sigma dW_t$. *Parameters*: $\kappa = 1.0$, $\alpha = 1.0$, $\sigma = 0.3$, $x_0 = 1.0$, $T = 1.0$, $n\_steps = 100$.

- **Cox-Ingersoll-Ross Process (CIR).** *Formula*: $dX_t = \kappa(\alpha - X_t)dt + \sigma\sqrt{X_t}dW_t$. *Parameters*: $\kappa = 0.5$, $\alpha = 1.0$, $\sigma = 0.3$, $x_0 = 1.0$, $T = 1.0$, $n\_steps = 100$.

- **Physical Stochastic Differential Equation (Phys).** *Formula*: $dX_t = (\eta - \nu/2)X_t^{2\eta-1}dt + X_t^{\eta}dW_t$. *Parameters*: $\eta = 0.7$, $\nu = 0.3$, $x_0 = 1.0$, $T = 1.0$, $n\_steps = 100$.

- **Nonlinear Stochastic Differential Equation (NL).** *Formula*: $dX_t = X_t(\kappa - (\sigma^2 - \kappa X_t))dt + \sigma X_t^{3/2}dW_t$. *Parameters*: $\kappa = 0.5$, $\sigma = 0.3$, $x_0 = 1.0$, $T = 1.0$, $n\_steps = 100$.

- **Power Law Volatility Model (PL).** *Formula*: $dX_t = \kappa(\alpha - X_t)dt + \sigma X_t^{p}dW_t$. *Parameters*: $\kappa = 0.5$, $\alpha = 1.0$, $\sigma = 0.3$, $p = 0.5$, $x_0 = 1.0$, $T = 1.0$, $n\_steps = 100$.

- **Polynomial Drift Model (Poly).** *Formula*: $dX_t = (\alpha_{-1}X_t^{-1} + \alpha_0 + \alpha_1 X_t + \alpha_2 X_t^2)dt + \sigma X_t^{3/2}dW_t$. *Parameters*: $\alpha_{-1} = -0.1$, $\alpha_0 = 0.1$, $\alpha_1 = 0.2$, $\alpha_2 = -0.05$, $\sigma = 0.3$, $x_0 = 1.0$, $T = 1.0$, $n\_steps = 100$.

Non-Gaussian processes incorporate discontinuous innovations through jump components or Lévy-driven noise, essential for modeling systems with abrupt state changes. Jump-diffusion models generalize Brownian motion through $dX_t = \mu X_t dt + \sigma X_t dW_t + J_t dN_t$, where compound Poisson jumps $J_t$ superimposed on diffusion dynamics capture financial market crashes and epidemiological surges. The generalized gamma process $dX_t = a(X_t)dt + b(X_t)dW_t + c(X_t)dN_t$ further couples state-dependent jump intensities with multiplicative noise, simulating information cascades in social networks through its self-exciting feedback mechanism. The detailed configurations are listed below.

- **Jump Diffusion Process (JUMP).** *Formula*: $dX_t = \mu X_t dt + \sigma X_t dW_t'$, where $dW_t' = dW_t + dJ_t$. *Parameters*: $\mu = 0.05$, $\sigma = 0.2$, $x_0 = 1.0$, $T = 1.0$, $n\_steps = 100$, $jump\_rate = 5.0$, $jump\_mean = -0.1$, $jump\_std = 0.8$

- Generalized **Gamma** Process (Gamma). *Formula*: $dX_t = a(X_t, t)dt + b(X_t, t)dW_t + c(X_t, t)dN_t$ (specific form: $dX_t = (a_{coef}X_t)dt + (b_{coef}X_t)dW_t + (c_{coef}X_t)dJ_t$). *Param-*

*eters*: $a_{coef} = 0.1$, $b_{coef} = 0.2$, $c_{coef} = 0.3$, $jump\_rate = 3.0$, $jump\_mean = 0.1$, $jump\_std = 0.2$, $x_0 = 1.0$, $T = 1.0$, $n\_steps = 100$.

**Distributional Configurations.** As summarized in Table 5, we consider both unimodal and multimodal distributions with varying tail behaviors and skewness.

| Distribution | Form | Parameters |
|---|---|---|
| Normal | $\mathcal{N}(\mu, \sigma^2)$ | $\mu = 0.0$, $\sigma = 1.0$ |
| Log-Normal | $\mathrm{LogNormal}(\mu, \sigma^2)$ | $\mu = 0.0$, $\sigma = 1.0$ |
| Student-$t$ | $t(df)$ | $df = 3.0$ |
| Chi-Squared | $\chi^2(df)$ | $df = 3.0$ |
| F-Distribution | $F(df_1, df_2)$ | $df_1 = 5.0$, $df_2 = 2.0$ |
| Logistic | $\mathrm{Logistic}(\mu, s)$ | $\mu = 0.0$, $s = 1.0$ |
| Cauchy | $\mathrm{Cauchy}(x_0, \gamma)$ | $x_0 = 0.0$, $\gamma = 1.0$ |
| Laplace | $\mathrm{Laplace}(\mu, b)$ | $\mu = 0.0$, $b = 1.0$ |
| Gamma | $\Gamma(k, \theta)$ | $k = 2.0$, $\theta = 2.0$ |
| Beta | $\mathrm{Beta}(\alpha, \beta)$ | $\alpha = 2.0$, $\beta = 5.0$ |
| Exponential | $\mathrm{Exp}(\lambda)$ | $\lambda = 1.0$ |
| Uniform | $\mathcal{U}(a, b)$ | $a = -1.0$, $b = 1.0$ |
| Gaussian Mixture (bimodal) | $\sum_{i=1}^{2} \pi_i \mathcal{N}(\mu_i, \sigma_i^2)$ | $\pi = [0.5, 0.5]$ $\mu = [-2, 2]$ $\sigma = [0.5, 0.5]$ |
| Gaussian Mixture (trimodal) | $\sum_{i=1}^{3} \pi_i \mathcal{N}(\mu_i, \sigma_i^2)$ | $\pi = [1/3, 1/3, 1/3]$ $\mu = [-3, 0, 3]$ $\sigma = [0.5, 0.7, 0.5]$ |
| Gaussian Mixture (quadrimodal) | $\sum_{i=1}^{4} \pi_i \mathcal{N}(\mu_i, \sigma_i^2)$ | $\pi = [0.25, 0.25, 0.25, 0.25]$ $\mu = [-4, -1.5, 1.5, 4]$ $\sigma = [0.5, 0.4, 0.4, 0.5]$ |

Table 5: Distributional configurations used in simulated experiments.

## MODEL CONFIGURATIONS

The drift network $\mu_\theta(X_t, t)$ and the diffusion network $\sigma_\phi(X_t, t)$ are both implemented as feedforward neural networks with three hidden layers [64, 128, 64], using the `SiLU` activation function. To ensure positivity of the output, the diffusion network $\sigma_\phi$ applies a `Softplus` activation function in its final layer. ITD module $\epsilon_\theta(\mathrm{d}t)$—adopts a hierarchical design. It consists of a base network with hidden layers [256, 512, 256], using a base dropout rate of 0.2. On top of this, our inference dropout uses $p_2 = 0.5$. For real-world sequential datasets, the drift and diffusion functions are parameterized by a 2-layer gated recurrent unit (GRU) with a hidden dimension of 128, enabling effective modeling of temporal dependencies in historical data. All networks employ `BatchNorm` to ensure training stability.

## OPTIMIZATION AND TRAINING

Latent SDEs with ITD are trained end-to-end by maximizing the evidence lower bound (ELBO), consistent with the original formulation. We employ the Adam optimizer with an initial learning rate of $1 \times 10^{-4}$. The learning rate is dynamically adjusted using a `ReduceLROnPlateau` scheduler, which halves the rate if the validation loss does not improve for 10 consecutive epochs. For synthetic datasets, training is performed for up to 40 epochs with a batch size of 1024. For real-world benchmarks, we use a batch size of 64 and train for up to 40 epochs with early stopping based on validation performance. All experiments are implemented in PyTorch and executed on NVIDIA A100 GPUs. For fair comparison, part of the results are directly taken from (Park et al., 2021). For all baseline models, we followed the original implementations and performed hyperparameter search over batch size and learning rate. Specifically, we considered batch sizes $\{32, 64, 128, 256, 512\}$ and learning rates $\{1 \times 10^{-5}, 5 \times 10^{-5}, 1 \times 10^{-4}, 5 \times 10^{-4}, 1 \times 10^{-3}\}$. The best configuration was

selected based on validation performance, with early stopping applied to prevent overfitting. We report the average performance over five independent runs with different random seeds, using the same hyperparameters selected via validation for each run.

## PROOFS

**Proof of Theorem 3.4.** Fix $x \in X$. By Assumption 3.3, the conditional distribution $P_{Y|X=x}$ admits a continuous quantile function $F_x^{-1} : (0,1)^d \to Y$ such that if $U \sim \mathrm{Unif}(0,1)^d$, then $F_x^{-1}(U) \sim P_{Y|X=x}$. Since $Y$ is bounded, uniform continuity of $F_x^{-1}$ follows.

Consider the dropout masks $M = (M^{(1)}, \ldots, M^{(L-1)})$ with $M^{(\ell)} \sim \mathrm{Ber}(p)^{n_\ell}$ independently, and define a deterministic mapping $\psi$ such that $Z = \psi(M) \in \mathcal{Z}$ takes values in a finite set $\mathcal{Z} = \{z_1, \ldots, z_K\}$ with probabilities $\{q_1, \ldots, q_K\}$ determined by $p$ and the layer widths. The law of the randomized network output can then be written as

$$\mathcal{L}(f_\theta(M,x)) = \sum_{k=1}^{K} q_k \, \delta_{f_\theta(z_k,x)}.$$

Let $\{u_k\}_{k=1}^K \subset (0,1)^d$ be a finite grid. By uniform continuity of $F_x^{-1}$, there exists such a grid satisfying

$$\sup_{u \in (0,1)^d} \inf_{k \le K} \|F_x^{-1}(u) - F_x^{-1}(u_k)\| < \varepsilon/2.$$

This implies that the discrete distribution $\sum_{k=1}^{K} q_k \, \delta_{F_x^{-1}(u_k)}$ approximates $P_{Y|X=x}$ in $\mathcal{W}_1$ distance up to $\varepsilon/2$.

By the universal approximation theorem for ReLU networks, there exists a network $f_\theta$ with sufficient depth and width such that

$$\sup_{x \in X} \sup_{k \le K} \|f_\theta(z_k,x) - F_x^{-1}(u_k)\| < \varepsilon/2.$$

It follows that

$$\mathcal{W}_1\big(\mathcal{L}(f_\theta(M,x)),\ P_{Y|X=x}\big) < \varepsilon \quad \text{uniformly for } x \in X.$$

Therefore, there exist $L$, widths $(n_\ell)$, dropout rate $p$, and weights $\theta$ such that the randomized network output distribution approximates the target conditional law within $\varepsilon$ in $\mathcal{W}_1$ distance, as claimed. $\qquad\square$

## MORE ANALYSIS

**Sensitivity Analysis on Dropout Probability.** We further examined the effect of varying the dropout probability $p$ during inference. As shown in Table 6, the results indicate that the performance is relatively insensitive to the exact choice of $p$, as long as the underlying neural network has sufficient capacity (i.e., enough parameters). This suggests that the expressive power of the network can compensate for different stochastic rates, ensuring stable behavior across a range of $p$ values.

| $p$ | Accuracy ($\uparrow$) | NLL ($\downarrow$) |
|---|---|---|
| 0.1 | 0.612 | 0.485 |
| 0.3 | 0.609 | 0.481 |
| 0.5 | 0.611 | 0.479 |
| 0.7 | 0.610 | 0.483 |
| 0.9 | 0.608 | 0.487 |

Table 6: Effect of varying inference-time dropout probability $p$ on Latent-SDE (with ITD), evaluated on the synthetic dataset. Results show predictive accuracy ($\uparrow$) and negative log-likelihood (NLL, $\downarrow$). Performance remains stable across different $p$ values when the network has sufficient capacity.

**Other Discretization Methods: Complexity Analysis.** As shown in Table 7, higher-order schemes such as the Milstein method and Enhanced Local (EL) methods can theoretically reduce the discretization bias of Euler–Maruyama. However, their computational and memory costs increase substantially: Milstein requires evaluating derivatives of the diffusion term, while EL methods rely on higher-order expansions whose cost grows with the order $k$. Consequently, these methods are difficult to scale to high-dimensional systems or long-horizon forecasting tasks. In contrast, our Inference-Time Dropout (ITD) maintains constant complexity while effectively mitigating discretization bias, providing a lightweight and scalable alternative.

| Method | Time Complexity | Space Complexity |
|---|---|---|
| Euler–Maruyama (baseline) | $O(1)$ per step | $O(1)$ |
| Milstein Scheme | $O(1 + C_{g'})$ per step | $O(1)$ |
| Enhanced Local (EL) | $O(k)$ per step | $O(k)$ |
| Inference-Time Dropout (ITD) | $O(1)$ per step | $O(1)$ |

Table 7: Time and space complexity of different discretization strategies.

