# OpenReview forum: "Mitigating Discretization Bias in Neural Stochastic Differential Equations via Inference-Time Dropout"
_ICLR.cc/2026/Conference — ICLR 2026 Conference Withdrawn Submission_

### Official Review · Reviewer_drzD · 2025-10-29

**Soundness:** 2
**Presentation:** 3
**Contribution:** 2
**Rating:** 4
**Confidence:** 4

**Summary:**

This paper addresses discretization bias in Neural Stochastic Differential Equations (NSDEs), particularly those using the Euler-Maruyama (EM) scheme. The authors argue this bias accumulates and harms inference performance. The paper proposes Inference-Time Dropout (ITD), a method that activates dropout during the inference phase.

**Strengths:**

- The paper clearly articulates the problem of discretization bias in NSDEs, an important and often-overlooked practical issue in the field. The proposed ITD method is lightweight, and requires no model retraining or architectural changes, giving it high potential for adoption by practitioners.

- The method demonstrates consistent and significant improvements on end-user metrics (e.g., MMD, NLL, MSE) across a variety of synthetic and real-world datasets.

- The paper provides a formal theorem (Theorem 3.4) establishing that dropout-induced randomness can universally approximate conditional output distributions, which is a non-trivial theoretical insight.

**Weaknesses:**

- The central claim that ITD mitigates discretization bias is not rigorously demonstrated. The provided theory (Theorem 3.4) concerns distributional expressivity, not SDE weak or strong convergence error. The paper must provide a direct theoretical link.

- The empirical results, while showing large improvements (e.g., in MMD), are presented without confidence intervals, error bars, or significance testing. Given the stochastic nature of dropout, this omission makes it difficult to assess the robustness and reproducibility of the results.

- Critical details of the ITD mechanism remain under-specified. The paper must clarify precisely how dropout masks are sampled (e.g., i.i.d. per step, fixed per path) and whether or how the injected noise is scaled to match diffusion scaling (e.g., proportional to $\sqrt{\Delta t}$).

**Questions:**

- Can you provide direct empirical evidence, such as a plot of weak error versus step size $\Delta t$, that demonstrates ITD reduces SDE discretization bias, rather than just improving downstream metrics?

- Can you provide a more formal derivation linking the variance from dropout to the reduction of EM bias, beyond the qualitative argument about expressivity (Theorem 3.4)?

- How is the injected noise $\epsilon_\theta(\Delta t)$ scaled with the time step $\Delta t$? Is its variance proportional to $\sqrt{\Delta t}$ to match the scaling of Brownian motion increments?

- What is the computational overhead of ITD? How many forward passes (samples) are required at inference to achieve the reported results, and how sensitive is performance to this number?

- The paper claims to improve uncertainty modeling but provides no standard calibration metrics. Can you provide results for Expected Calibration Error (ECE) or other uncertainty quantification measure?

---

### Official Review · Reviewer_tWgP · 2025-10-30

**Soundness:** 2
**Presentation:** 2
**Contribution:** 2
**Rating:** 4
**Confidence:** 3

**Summary:**

This paper investigates the problem of discretization bias in Neural Stochastic Differential Equations (NSDEs) — a systematic error introduced when continuous-time SDEs are numerically approximated via the Euler–Maruyama (EM) scheme. To address this issue, the paper proposes Inference-Time Dropout (ITD) — a simple yet effective mechanism that reinterprets dropout as a stochastic correction layer at inference. Instead of using Gaussian noise increments, the model replaces them with neural-network-generated random vectors with fixed dropout masks. This approach effectively allows the network to learn and sample more complex, non-Gaussian noise structures, potentially compensating for EM-induced bias.

**Strengths:**

ITD is lightweight, easy to integrate into existing NSDE frameworks, and does not significantly alter model complexity or training pipelines. Empirical evaluation on synthetic and real-world datasets shows consistent improvements in metrics.

**Weaknesses:**

1. Limited theoretical novelty.
  * Theorem 3.1 restates a classical result on EM convergence (see [1]) and should be properly cited rather than presented as new.
  * Theorem 3.4 follows directly from the universal approximation theorem for ReLU networks; its contribution to the novelty is limited.
2. Lack of analysis of improvement.
Although empirical results show strong gains, the paper does not isolate why ITD helps:
  * Is the improvement due to mitigating discretization bias specifically?
  * Or does the additional noise simply improve optimization or model expressivity?
3. Insufficient exploration of related directions.
The connection to Mixture-of-Experts (MoE) and Gaussian Mixtures is conceptually intriguing but lacks empirical substantiation. No experiments explicitly test these analogies.
4. Incomplete baselines.
Many comparisons are made against earlier NSDE and ODE models, while recent developments in neural SDEs and diffusion processes are missing.
Recommended additional baselines:
  * Higher-order or adaptive discretization methods from Table 7.
  * Modern neural architectures: ContiFormer (2024), Stable Neural SDEs [2, 3].
  * Non-Brownian or fractional noise models [4–6].

  ITD introduces additional trainable components; therefore, comparisons should include the number of parameters and training and inference time to properly estimate the tradeoff between quality and complexity of the approach.

5. Ambiguity in discretization bias mitigation.
Despite claims that ITD reduces discretization bias, no quantitative metric of bias reduction (e.g., error vs. step size) is presented. Improvements in prediction and uncertainty metrics might reflect better fitting capacity rather than bias correction per se.
6. Details on the MNIST experiment (Figure 3) are missing.

[1] https://urbain.vaes.uk/static/teaching/lectures/build/lectures-w6.pdf

[2] ContiFormer: Continuous-Time Transformer for Irregular Time Series Modeling

[3] Stable Neural Stochastic Differential Equations in Analyzing Irregular Time Series Data

[4] Learning Fractional White Noises in Neural Stochastic Differential Equations

[5] Fractional SDE-Net: Generation of Time Series Data with Long-term Memory

[6] Variational Inference for SDEs Driven by Fractional Noise

**Questions:**

1. Does the improvement persist when drift and diffusion are fixed to ground-truth dynamics?
2. How does ITD behave under very small time steps (Δ → 0)? Does the noise distribution converge to Brownian increments, or does the model maintain a persistent bias?
3. What is the computational overhead of ITD during training and inference?

---

### Official Review · Reviewer_tFtT · 2025-10-31

**Soundness:** 1
**Presentation:** 1
**Contribution:** 2
**Rating:** 2
**Confidence:** 4

**Summary:**

This work presents an alternative approach to simulate from a Neural Stochastic Differential Equation. Instead of an Euler simulation, the proposed approach replaces the iid Gaussian increment with an increment output by neural networks with dropout enabled at inference time. An analysis of the error of the Euler simulation is presented, along with a distribution-wise approximation theorem for neural networks with dropout.

**Strengths:**

Theorem 3.4, if true, is an interesting observation that neural networks can not only approximate a deterministic map but also approximate a conditional distribution by allowing some components to be random. This is a novel contribution to me.

**Weaknesses:**

While the central theorem (Thm 3.4) does seem to be interesting, I don't think its impact and applicability are properly conveyed by applying it to NSDE.

1. This work does not offer any guarantee that the incremental distribution output by the IDT will lead to a Brownian motion $W_t$.  Hence, the processes generated by NSDE with IDT will potentially not have the law of an Ito's SDE. If the authors would like to show that IDT can outperform Euler, I'd say a bound similar to Theorem 3.1, but for IDT with a better rate, would be necessary. Overall, the process driven by IDT should be $$X_t = f(X_t, t) d t + g(X_t, t) d Y_t$$ where $Y_t$ is another process directly approximated by IDT. Yet, this is beyond the scope of Ito's SDE, and I believe it has the name "Rough Stochastic Differential Equations".




2. This work also misses a lot of critical citations. Here, I provide only a few examples, rather than an exhaustive list. For example, in the proof "the universal approximation theorem for ReLU networks" is used but never stated, proved, or cited. There are also no citations provided for the datasets. There is a question mark around line 165 indicating a missing citation. Terms like DGP, maximum mean discrepancy (MMD), and jump processes are not defined or cited. Moreover, some citations seem to be mismatched with the paper. For example, around line 253, "...GAN-SDE (Li et al., 2020),", but as far as I can understand, Li et al., 2020 is not on GAN-SDE but proposes a generalization of the adjoint method to NSDE to reduce the memory footprint of NSDE backpropagation.




3. Some recent works in NSDE are not covered in the literature review. On top of my head, at least [1, 2, 3, 4] (not an exhaustive list) are missing.




4. The methods applied in experiments are too old. As far as I'm aware, the [2, 3, 5, 6] are considered SOTA (either in performance or in efficiency) and open-sourced, and there is an implementation of [4] provided in [6]. Also, given that the paper is trying to reduce the numerical error, an interesting baseline would be higher-order solvers for SDEs like Milstein.




5. There is no information on the real-world datasets used in experiments. From my perspective, Beijing Air Quality is not an appropriate choice for NSDE benchmark because it only has a length of 24 according to [5]. The horizon is too short to test the capacity of modern deep learning models. (In case the authors are looking for better datasets, the stock prices are publicly available, can have a very long horizon if you sample frequently, and they're known to be best modeled by SDEs.)




6. It is not clear to me how NSDE, presented in the form of this paper, applies to reconstruction and prediction (while neither  task is precisely defined in the paper), and why MSE and NLL are reasonable measures of performance. NSDE is a generative model for temporal data and matches the distribution of the training samples. Thus, I think it would be more reasonable to make distribution-wise comparisons than pathwise comparisons, $i.e.$, it would be more reasonable to compare the law of samples generated by NSDE against the law of real data, or at least the marginal distribution of generated samples vs real data.






## References:
[1]  Liu, Y.-J., Lu, M., Nock, M. K., & Yacoby, Y. (2025). Neural stochastic differential equations on compact state-spaces. arXiv preprint arXiv:2508.17090. https://arxiv.org/abs/2508.17090

[2] Zhang, J., Viktorov, J., Jung, D., & Pitler, E. (2025). Efficient training of neural stochastic differential equations by matching finite dimensional distributions. arXiv preprint arXiv:2410.03973. https://arxiv.org/abs/2410.03973

[3] Snow, L., & Krishnamurthy, V. (2025). Efficient neural SDE training using Wiener-space cubature. arXiv preprint arXiv:2502.12395. https://arxiv.org/abs/2502.12395

[4]  Bonnier, P., & Oberhauser, H. (2024). Proper scoring rules, gradients, divergences, and entropies for paths and time series. Bayesian Analysis, 1-32. https://doi.org/10.1214/24-BA1435

[5]  Kidger, P., Foster, J., Li, X., Oberhauser, H., & Lyons, T. (2021). Neural SDEs as infinite-dimensional GANs. arXiv preprint arXiv:2102.03657. https://arxiv.org/abs/2102.03657

[6] Issa et al. (2023) Issa, Z., Horvath, B., Lemercier, M., & Salvi, C. (2023). Non-adversarial training of neural SDEs with signature kernel scores. In Advances in neural information processing systems (Vol. 36, pp. 11102–11126).

**Questions:**

see weakness

---

### Official Review · Reviewer_K1EV · 2025-10-31

**Soundness:** 2
**Presentation:** 1
**Contribution:** 2
**Rating:** 2
**Confidence:** 4

**Summary:**

This paper propose a novel inference-time mechanism, Inference-Time Dropout (ITD), designed as a data-driven stochastic correction to reduce discretization bias. Instead of relying on the standard Gaussian increment, ITD replaces it with a learned noise process parameterized by a neural network with fixed dropout masks. Theoretical analysis (Theorem 3.4) claims ITD can act as a universal conditional distribution approximator, while empirical studies across synthetic and real-world datasets suggest performance improvements compared to standard EM-based Neural SDEs.

**Strengths:**

1. **Relevance**: The paper highlights an important and under-discussed issue, discretization bias, within Neural SDE frameworks. Although this bias is well-known in numerical analysis, its practical impact in Neural SDEs has been rarely studied.

2. **Empirical performance**: Empirical results show that ITD-based models outperform baselines in certain scenarios. This suggests that the learned noise can capture distributional properties that conventional EM solvers miss.

3. **Conceptual novelty**: It seems that the idea of learning a non-Gaussian, structured noise process at inference time is novely.

**Weaknesses:**

1. **Unclear mechanism of ITD**:

- The paper immediately introduces Inference-Time Dropout (ITD) but does not clearly explain its mechanism. It states that dropout masks are maintained during inference, yet it remains unclear how dropout is actually applied whether it is implemented within the drift and/or diffusion networks of the Neural SDE.
- The definition of the key component $\epsilon_\theta$ is vague.


2. **Core conceptual disconnect**:
The core message of the paper is ambiguous. While ITD claims to mitigate discretization bias, the main content does not provide a clear explanation or theoretical basis for such an effect.  The fact that the learned noise process $\epsilon_\theta$ can model an arbitrary noise distribution is not equivalent to reducing discretization bias. In particular, Theorem 3.4 merely discusses the universal approximation property of ReLU networks, which is conceptually disconnected from the claimed contribution of ITD.

3. **Thereotical irrelevance of Theorem 3.4**: Theorem 3.4 only shows that dropout-based networks can approximate conditional distributions under Wasserstein distance, which is unrelated to discretization bias or convergence rates in SDE solvers. I believe what is required is a theorem showing that ITD yields a better weak/strong order of accuracy compared to EM.

4. **Inappropriate experimenta ldesign**:
- To validate “bias mitigation,” experiments should show how error scales with step size $\Delta$. The paper uses a fixed $\Delta$ compares model performance, making it impossible to separate bias reduction from mere overparameterization or regularization effects.

- It is unlear why the MNIST generate experiment is related to the purpose of this work.

5. Thereom 3.1 is a well-established standard fact. I think citing a proper reference is enough.

**Questions:**

Please see the weaknesses and the following questions:

1. How dropout is actually applied? Is it implemented within the drift and/or dfifusion networks of the Neura SDE?

2. Could you specify how $\epsilon_\theta$ is formulated and how the parameters $\theta$ are trained

3. Please report the performance with respect to varying step size.

---

### Note · Authors · 2025-11-25

**Comment:**

We sincerely thank the reviewers for the time, effort, and valuable feedback provided during the evaluation of our manuscript. Your comments and suggestions have been truly helpful for our research. We will further improve and strengthen our work based on these insights.

**Withdrawal Confirmation:**

I have read and agree with the venue's withdrawal policy on behalf of myself and my co-authors.